# msf-CNN: Multi-Stage Fusion with Convolutional Neural Networks for TinyML

## Abstract

AI spans from large language models to tiny models running on microcontrollers (MCUs). Extremely memory-efficient model architectures are decisive to fit within an MCU's tiny memory budget e.g., 128kB of RAM. However, inference latency must remain small to fit real-time constraints. An approach to tackle this is *fusion*, which aims to optimize data flows across neural network layers. In this paper, we introduce *msf-CNN*, a novel technique that efficiently finds optimal fusion settings for convolutional neural networks (CNNs) by walking through the fusion solution space represented as a directed acyclic graph. Compared to previous work on CNN fusion for MCUs, msf-CNN identifies a wider set of solutions. We published an implementation of msf-CNN running on various microcontrollers (ARM Cortex-M, RISC-V, ESP32). We show for instance that msf-CNN achieves inference using 50% less RAM compared to the prior art (MCUNetV2 and StreamNet). msf-CNN thus offers additional flexibility for system designers.

## 1. Introduction

Artificial Intelligence of Things (AIoT) is a domain aiming to embed AI in the smallest networked devices (Ghosh et al., 2018). As such AIoT is pushing the miniaturization of Deep Neural Networks (DNNs) to fit microcontroller-based hardware, which enables various applications at the edge of the network. Use-cases include vision/audio recognition, environmental monitoring, personalized medical care, etc. However, imbalance between the increasing resource requirements of DNNs and the very limited computation capacity (CPU in MHz) and memory resource of Microcontroller Units (MCUs) remains a challenge in deploying DNNs on Internet of Things (IoT) devices. For instance, as described in RFC7228 (Bormann et al., 2014), billions of IoT devices are resource-constrained devices, with Random Access Memory (RAM) smaller than 50 KiB, and Flash memory smaller than 250 KiB. On the other hand, even a single convolutional layer in quantized ResNet-34 (Koonce, 2021; He et al., 2016) consumes around 414.72 KiB in RAM. This example highlights the huge gap between memory budgets on IoT devices and RAM usage of DNNs.

A technique aimed at decreasing this gap is layer *fusion*, introduced in (Alwani et al., 2016). Initially targeting FPGAs, fusion reduces off-chip Dynamic RAM (DRAM) requirements and communication bus transfer costs for inference with CNNs. Fusion is great for low-memory devices because it can save up to 95% of RAM usage. Moreover, Fusion decouples input size from memory usage, allowing for larger input. Recent work has thus explored the use of fusion on MCUs, for example, to improve the memory consumption of the first few convolutional layers of MobileNetV2 (Lin et al., 2021).

Nevertheless, we observe that significant issues linger on MCUs. First, intermediate feature maps inside the fusing block incur a high (re)compute cost. Second, input size limits hamper many use-cases such as medical image processing, sequence time series analysis (e.g. audio application), etc. Third, implementations of fusion on MCUs have so far been very hardware-specific (e.g. bound to the ARM-Cortex-M7 instruction set) and model-specific (e.g. bound to CNN mobile inverted blocks).

**Contributions –** With the goal of improving on the above issues, we report on following work:

- We propose msf-CNN, a fusion-based approach to achieve ultra-low RAM footprint of neural network inference and we open-source its implementation;

- We formulate the problem of finding optimal fusion settings that minimize peak RAM usage or compute cost of neural networks as a variant shortest path problem.

- We provide graph models representing multi-stage fusion neural networks, which encode peak RAM usage and compute cost of single and fused layers.

- We designed a pruning strategy to squeeze the search space and use graph-based algorithm to find solutions in reasonable time complexity (From $O(2^{N-2})$ to $O(N^2)$).

[1]. Correspondence to: Anonymous Author .

Preliminary work. Under review by the International Conference on Machine Learning (ICML). Do not distribute.

- We improved global pooling and dense operators to further squeeze RAM usage without compute overhead.

- We released preliminary evaluation results on MCU-based IoT boards. We compared common CNN, StreamNet, MCUNetV2 and msf-CNN on a variety of microcontrollers. We show that msf-CNN allows new trade-off between memory saving and compute overhead.

## 2. Background

**Fusion for DNN on FPGA & GPU –** Fusion was initially proposed in (Alwani et al., 2016) as a fusion scheme for Convolutional Neural Network (CNN) deployed on Field Programmable Gate Array (FPGA) to reduce the off-chip DRAM usage and I/O overhead. Instead of computing the complete feature maps for each layer, it fuses convolutional layers into a single block (pyramid structure) and computes only one or a few output elements. This approach requires only small portions (tiles) of the feature maps loaded onto DRAM. However, the reduction of RAM is at the cost of re-computing the overlapped elements in feature maps required by adjacent fused layers. DeFiNES (Mei et al., 2023), another fusion framework, explored different cache strategies within fused layers to alleviate the re-computation issue. (Fully-recompute, H-Cached & V-recompute, and Fully-cache). Fully-recompute eliminates caching entirely, requiring all overlapping input tensor elements to be recalculated; H-cached & V-recompute caches elements along the horizontal axis while recomputing vertical overlaps; and Fully-cache retains all overlapping elements in memory. These approaches illustrate a critical trade-off—enhanced caching progressively reduces compute redundancy but proportionally increases RAM usage, with cached element quantity inversely correlating to compute overhead and directly scaling with memory demands. Additional work has also applied fusion on GPUs, for instance (Pinckaers et al., 2022) used it for cancer detection in medical pictures.

**Fusion on MCUs –** Work on MCUNetV2 (Lin et al., 2021) has applied fusion on MobileNetV2 to reduce the peak RAM usage. It revealed that layers at the head of the model dominate the RAM usage. Hence these layers were fused into one block to reduce RAM usage significantly. The recompute issue was mitigated by redistributing the receptive field, so the receptive field inside the fusion block was decreased and regained at a later stage. Work on StreamNet (Zheng et al., 2024a) introduced a two-dimensional tensor cache to significantly reduce re-compute operations in a fusion block and applied brute force to search for optimal fusion position and cache depth. Nevertheless, no prior work explored the potential of multiple fusion blocks in CNNs.

**Representing DNNs as Inverted Dataflow Graphs –** Dataflow graph have been widely used for modeling DNN,

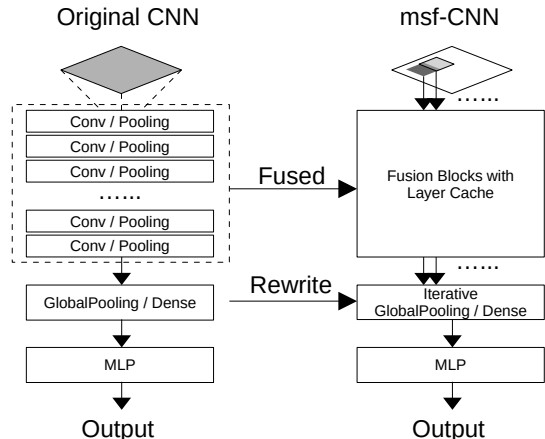

*Figure 1.* Overview of msf-CNN. The convolutional layers are fused into several fusion blocks based on the optimal setting found by optimizer. We let global Pooling and dense layers compute the outputs iteratively to further squeeze RAM usage.

as pioneered by TensorFlow and PyTorch (Abadi et al., 2016; Paszke et al., 2019). The data (tensor) flows alongside the directed edge between nodes which indicates the operations (convolution, pooling, addition, etc.) applied on the incoming edges (tensors). This representation shows the producer-consumer relations among operations and has great expressiveness and flexibility, enabling automatic differentiation and concurrent execution of independent operations.

## 3. High-Level Idea

Inspired by the above previous works, msf-CNN aims to answer the following questions: **(1)** Where to fuse and how to determine the fusion position/depth? **(2)** Under specific resource constraints, how to find the optimal fusion settings?

As depicted in Figure 1, msf-CNN determines fusion settings (fusion position and depth), transforms layers accordingly into fusion blocks and rewrites global pooling and dense layers as their iterative implementation, which can further squeeze RAM usage without any computation overhead.

To guide us in doing so, we use *inverted* dataflow graphs to model CNNs, where tensors are represented as nodes, and operations are depicted as edges connecting them. On this graph, we encode into the edges the resource usage of the operations, and use additional edges to represent fusion blocks. This allows us to design graph-based strategies to find optimal solutions with lower computational complexity using proven graph algorithms.

## 4. Problem Definitions & Assumptions

We aim to solve a pair of dual optimization problems. Let $\chi$ be the set of all possible configurations for fusion blocks. We **define *P1* as the problem of minimizing peak RAM usage** subject to a computation cost limit:

$$\min_S P(\chi, S) \tag{1}$$

$$\text{s.t. } F(\chi, S) < F_{max} \tag{2}$$

where $P$ is the peak RAM usage, and $F$ is the computation overhead for inference under fusion setting $S$, relatively to inference without fusion (thereafter denoted *vanilla*). The compute cost limit and RAM limit are annotated by $F_{max}$ and $P_{max}$, respectively. Dually, we **define *P2* as the problem of minimizing computation cost** subject to a RAM footprint limit:

$$\min_S F(\chi, S) \tag{3}$$

$$\text{s.t. } P(\chi, S) < P_{max} \tag{4}$$

Without loss of generality, we only discuss fusion blocks of convolutions. We assume a *H-Cache* scheme, which we chose to be a good trade-off between buffer size and recompute cost on MCUs. For a fusion block containing $n$ layers, the cache buffer size of the $i$-th layer under H-cache scheme is given by

$$\text{Buf}_i = t_i \times k_i \times c_i^{in} \tag{5}$$

where $t_i$, $k_i$ and $c_i^{in}$ are the tile size, kernel size and input channels number, respectively. Obviously, the first layer of the fusion block does not need any input cache, thus $\text{Buf}_1 = 0$. The total cache size of the fusion block is $\text{Buf} = \sum_i \text{Buf}_i$.

In Appendix A, we further detail the analysis of the number of multiply–accumulate (MAC) operations.

## 5. DNN Graph Representation & Formulation

We interpret the optimization problems described in Section 4 by modeling the DNN as data-nodes graph. We transform the problem as a shortest path problem (Sedgewick, 2001) and use off-the-shelf graph algorithms to find a solution that minimizes the peak memory usage as well as compute cost during inference regarding specified constraints.

### 5.1. DNN Representation

As described in Section 2, we model a DNN as a directed acyclic graph (DAG) $G = (V, E)$ with data nodes $v_0, \ldots, v_n$ representing input/output tensors of consecutive layers and $m$ edges $e_1, \ldots, e_m$ that represent single layers or fusion blocks. Each edge is also encoded with resource

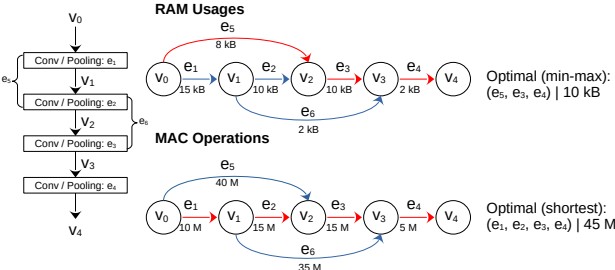

*Figure 2.* The neural network is modeled as a DAG. Nodes $v_n$ denote the tensors that are produced and consumed by the operators or possible fusion blocks. Edges $e_1, \ldots, e_4$ represent individual operators, while edge $e_5, e_6$ represent two candidate fusion blocks. Edges are annotated with the RAM usage and MAC amounts of their corresponding operators and fusion blocks.

requirements by layer or fusion block. Specifically, the first ($v_0$) and the last node ($v_n$) are the input and output tensor of the neural network, respectively.

In general, the edge represents the input/output relation of nodes and also indicates the fusion depth inside the neural network. For example, an edge that connects consecutive vertices $e = v_n \rightarrow v_{n+1}$ is a single layer that consumes $v_n$ as input tensor and outputs tensor $v_{n+1}$, while an edge that jumps over multiple vertices $e = v_n \rightarrow v_{n+m}, m > 1$ represents a fusion block with $m$ layers. Each **complete compute path** from $v_0$ to $v_n$ represents a fusion setting $S$.

A typical example depicted in Figure 2 explains how to use DAG for representing a simple neural network. Tensors are transformed into nodes, operators and fusion blocks are edges. Edges are encoded with RAM usages and MAC amounts of their corresponding operators. Hence, the problem is transformed to find an optimal path from the input node to the output node of the graph.

### 5.2. Encoding RAM Usage

We first calculate the RAM usages $P_{e_i}$ of all single layers and all possible fusion blocks inside the neural network by

$$P_{e_i} = I + O + Buf \tag{6}$$

where $I$ and $O$ are the size of input and output tensor, respectively. $Buf$ represents the cache buffer size of the fusion block, which is determined by the chosen cache scheme. In this work it is given in Equation (5). Trivially, for non-fused layers $Buf$ is always set to zero since no fusion cache is needed.

Thereafter, the calculated RAM usages are attached to the corresponding edges for further analysis. For a complete compute path contains $n$ edges $S = (e_{i_1}, \ldots, e_{i_n})$ we can

then calculate the overall peak RAM usage $P_S$ by

$$P_S = \max_{j=1...n} P_{e_{i_j}} \qquad (7)$$

### 5.3. Encoding Compute Cost

The encoding steps of compute cost are similar to encoding peak memory usage. Here we use MAC operations as the indicator of compute cost. In this paper, the MAC amount of fusion block is given in Equation (16) and Equation (17).

After attaching the calculated MACs to the edges, the total compute cost of a complete compute path $S$ is

$$C_S = \sum_{j=1}^{n} C_{e_{i_j}} \qquad (8)$$

Therefore, the **compute overhead factor** $F$ representing the ratio of the MAC amount after fusion to the vanilla, common one without fusion is expressed as

$$F = C_S / C_{vanilla}. \qquad (9)$$

For the constraints in Equation (2), users can set a maximum compute overhead factor $F_{max}$ expressed as

$$F_{max} = C_{max} / C_{vanilla}. \qquad (10)$$

In the following sections, we will discuss several graph-based algorithms to solve the optimization problem.

## 6. Searching for Optimal Fusion Settings

After building an inverted dataflow graph of a DNN with all possible fusion combinations (edges), the two dual problems are indeed transformed into classic graph problems: finding an optimal complete compute path from the input tensor node $v_1$ to the output tensor node $v_n$ under specific constraints.

**Impact of Search Space Size –** If we consider the unconstrained optimization, the solution is trivial: the single-source-single-target shortest path, which can be found by classical graph algorithm like Dijkstra's (Dijkstra, 2022) with the time complexity of $\mathcal{O}(E \log(V))$. However, when considering the constraints, it is necessary first to explore all possible complete compute paths that meet the conditions, which can potentially explode the complexity to $\mathcal{O}(2^{V-2})$ (Robert, 2002) in the worst case. Hence, we need a smarter strategy to squeeze the search space and avoid horrendous complexity.

### 6.1. Problem P1: Minimizing Peak RAM Usage

The unconstrained optimization is to find a complete compute path with minimal peak RAM usage, which is equivalent to finding the path that minimizes the maximum weight

of edges (minimax path problem). As mentioned above, this can be solved by modified Dijkstra algorithm. An example path with minimal peak RAM usage is presented in Figure 2.

For the constraint of compute cost limit (Equation (2)), the pruning strategy needs co-design with its optimization problem (Equation (1)). We noticed that all possible peak RAM usages have already been encoded into the edges. Therefore, the problem can be transformed into the following: we first construct a candidate solution set **candidate set** $\{S_0, S_1, \ldots, S_i, \ldots\}$ with

$$S_i = \arg\min_S C(G_i, S), \qquad (11)$$

$G_i :=$ subgraph of $G_{i-1}$, obtained by removing

all edges in $G_{i-1}$ with the maximal RAM usage, $\quad$ (12)

$$G_0 = G \qquad (13)$$

where $C(G_i, S)$ is the MAC amount of fusion setting $S$ in graph $G_i$. The candidate solution $S_i$ can be obtained by applying the shortest path algorithm. We then filter the candidate solutions to find those that satisfy the constraints and select the one with the smallest RAM usage as the optimal solution.

In this way, we avoid constructing a search space with a complexity of $\mathcal{O}(2^{V-2})$. Instead, we iteratively eliminate subgraphs and solve for candidate solutions, reducing the complexity to $\mathcal{O}(V^2)$. For most deep neural networks running on MCUs, this process can be done in few seconds.

### 6.2. Problem P2: Minimizing Compute Cost

We first discuss the unconstrained variant, which is identical to $P_{max} = \infty$. In this case, finding the solution is equivalent to finding the shortest complete compute path – the path with a minimal sum of MAC – of the graph, which can be again solved by classical algorithm like Dijkstra's (Dijkstra, 2022). Figure 2 shows an example with an optimal path marked in red.

When bringing back the constraint of RAM limit, the pruning step is simple: eliminating all edges with encoded RAM usage exceeding the limit. So, all paths in the graph will automatically fulfill the limitation.

### 6.3. Analytical Results

To explore the capability of these two dual optimizers, here we choose three variants of MobileNetV2 and MCUNet (Sandler et al.; Lin et al., 2021) with different scales for the pilot study: MobileNetV2 with width multiplier 0.35 and input size of $144 \times 144 \times 3$ (MBV2-w0.35), MCUNetV2-VVW-5fps with input size of $80 \times 80 \times 3$ (MN2-vvw5), MCUNetV2-320KB-ImageNet with input size of $176 \times 176 \times 3$ (MN2-320K). For optimizer of minimizing peak

RAM usage, the maximal compute overhead factor ranges from 1.1 to 1.5 then jumps to Infinite, which represents an unconstrained optimization. For optimizer of minimizing compute cost, the maximal peak RAM usage was set from 16 kB to 256 kB where each level represents a popular RAM capacity of mainstream MCUs.

As shown in Table 1, both optimizers can indeed theoretically suppress the peak RAM usage without violating all preset constraints. The high RAM usage compression is achieved with increase of deep fusion blocks, thereby introducing a high compute overhead. The extreme cases lay on the unconstrained optimization minimizing the RAM usage by more than 90%, while reluctantly introducing $1.6\times$ to $2.7\times$ of compute overhead. This is only suitable for time-intensive applications with a high limited RAM budget.

On the other hand, setting appropriate constraints can still lead to well-optimized configurations, with our tools offering flexibility to accommodate real-life scenarios. Under different thresholds on compute overhead factor or peak RAM usage, the solutions that optimizer found are all fulfill the constraints and with RAM usage all lower than the vanilla, un-fused setting. In some cases, it is even possible to compress RAM usage without incurring additional computational overhead. These pilot studies demonstrate the effectiveness of finding usable solutions under real-life constraints.

The analytical results were further validated by on-board experiments presented in Section 8.

## 7. msf-CNN Implementation Details

We have implemented the msf-CNN fusion mechanism on top of microTVM v0.16.0 (Chen et al., 2018). We use the TVM frontend to convert models into intermediate representation (IR), and rewrite the compute graph and low-level routines of operators to fit the fusion settings. We leveraged RIOT-ML (Huang et al., 2024) to benchmark the fused models (transform into C code by microTVM) on the IoT boards shown in Table 2.

**Sequential RAM Usage –** We have optimized the RAM usage of the global pooling and fully connected (Dense) layers. We observed that the outputs of these two basic blocks can be computed iteratively, and in most scenarios, their input dimensions are much larger than their output dimensions. As a result, we can temporally divide the input and sequentially process it through the iterative global pooling or dense layers, which further minimizes memory usage. If their upstream is a fusion block, this perfectly matches the feature of temporally split inputs, enabling them to be fused seamlessly.

**Iterative Computation of Global Pooling –** As illustrated

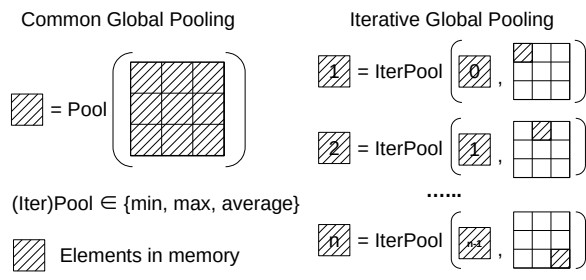

Figure 3. Comparison of common and iterative global pooling.

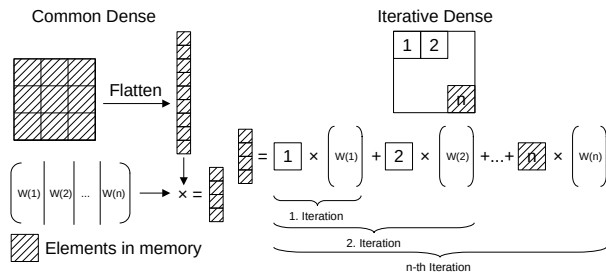

Figure 4. Comparison of common and iterative dense layer. The columns of the weight matrix are denoted as $w(n)$.

in Figure 3, standard global pooling requires that all elements of the input tensor stored in RAM. In our approach, the global pooling layer receives one or a few input elements at each step and iteratively updates the result. For a $7 \times 7$ global pooling layer, this allows us to compress the RAM usage to 2% of the original size, without introducing any redundant computations or computation overhead.

**Iterative Computation of Dense Layer –** We noted that the matrix multiplication in dense layers can be implemented by splitting the input vector into individual elements, multiplying each element with its corresponding weight column, and iteratively summing the results, as shown in Figure 4. Unlike the original approach, which requires the complete input tensor, this method processes only one element of the input tensor per iteration. For a $1024{\rightarrow}256$ dense layer, this approach compresses memory usage to 20% of the original.

## 8. Experiments on Microcontrollers

In this section we report on experiments running msf-CNN on various MCUs, aiming to validate both the correctness of our optimization strategies and their versatility when applied on diverse IoT devices.

Table 1. Analytical results with msf-CNN under different constraints. Vanilla: un-fused models. SAA: Same as above.

| | Constraint | MBV2-w0.35 | | MN2-vww5 | | MN2-320K | |
|---|---|---|---|---|---|---|---|
| | | RAM (kB) | $F$ | RAM (kB) | $F$ | RAM (kB) | $F$ |
| Vanilla | - | 194.44 | 1 | 96 | 1 | 309.76 | 1 |
| P1: $F_{max}$ | 1.1 | 67.905 | 1.1 | 32.792 | 1.04 | 190.096 | 1.04 |
| | 1.2 | (SAA) | | 26.128 | 1.11 | 186.736 | 1.19 |
| | 1.3 | 21.288 | 1.3 | 17.76 | 1.3 | 186.032 | 1.25 |
| | 1.4 | 15.34 | 1.38 | 13.376 | 1.35 | 156.672 | 1.37 |
| | 1.5 | (SAA) | | (SAA) | | 94.184 | 1.45 |
| | Inf | 7.887 | 1.68 | 12 | 1.96 | 42.643 | 2.69 |
| P2: $P_{max}$ | 16 kB | 15.34 | 1.38 | 13.376 | 1.35 | (No Solution) | |
| | 32 kB | 25.674 | 1.25 | 26.128 | 1.11 | | |
| | 64 kB | 63.741 | 1.23 | 38.576 | 1.02 | 62.88 | 2.02 |
| | 128 kB | 83.065 | 1.02 | 89.6 | 1 | 94.184 | 1.45 |
| | 256 kB | 181.44 | 1 | (SAA) | | 247.808 | 1 |

More concretely, we measured peak RAM usage and compute latency based on the fusion settings in Section 6.3, as reported in the following. As shown in Table 2, we carried out our experiments on the relevant 32-bit microcontroller architectures: Arm Cortex-M, Espressif Xtensa, and RISC-V. For our model zoo, we chose MBV2-w0.35, MN2-vww5 and MN2-320K as they are good representatives of backbones for applications in AIoT (Saha et al., 2022), as also used in prior works (Lin et al., 2021; Zheng et al., 2024a). We compare msf-CNN performance to the closest related work: MCUNetV2 (Lin et al., 2021) and StreamNet-2D (Zheng et al., 2024a), more simply denoted StreamNet in the following.

### 8.1. Minimal Peak RAM Usage

First, we evaluated solutions to P1 while relaxing Equation (2), i.e. the fusion settings with minimum peak RAM usage, without compute time constraint. Results are shown in Table 3. We observe that, compared to prior art (StreamNet-2D and MCUNetV2), msf-CNN can further reduce the peak RAM usage by 65% to 87%. We could even deploy the MBV2-w0.35 model onto the SiFive board that provides only 16 kB RAM (!). However, achieving this high compression ratio comes at the expense of increased computational latency, which we measured in Table 4. Interestingly, while clock frequency plays a decisive role, MCU architecture can also have a crucial effect, for larger models. For instance, notice latency with Xtensa esp32s3 at 240MHz *versus* RISC-V esp32c3 at 160 MHz, for the MN2-320K model (in Table 4). Nevertheless, we measured that latency increases $2\times$ to $5\times$ compared to vanilla (non-fused) CNN. Hence, such minimal RAM settings are only suitable for latency-tolerant applications on the smallest devices.

### 8.2. Impact of RAM Budget Limit

As shown in Table 5, the measured peak RAM usage consistently obeys to the given constraints, thereby validating the correctness of the optimizer and corroborating our analytical results. Based on these, we observe that higher RAM budgets result in shorter compute latency for the optimal fusion configurations identified by msf-CNN. This is because the optimizer tends to favor configurations with either no fusion or shallow fusion depths, which correspond to higher peak RAM usage but lower computational costs.

For the MBV2-w0.35 and MN2-vww5 models, our method outperforms MCUNetV2 when the RAM limit is set to 32kB and 64kB. Although our method does not surpass StreamNet-2D across the board, msf-CNN does demonstrate its flexibility, enabling users to select the optimal fusion configuration under varying memory budgets.

### 8.3. Impact of Computation Cost Limit

When capping computation cost as a constraint, the relation between compute latency and peak RAM usage is consistent (dual) with the previous section, such that higher compute overhead budgets result in longer compute latency and smaller peak RAM usage. We also observe that the ratio $F$ measuring the overhead compared to vanilla CNN (no fusion) is bigger than the $F_{max}$ we set for. This discrepancy comes from the fact that the optimizer computes the amount of MAC operations, whereas the full latency includes not only MAC operations but also I/O delays. In mainstream MCUs, model weights are stored in Flash rather than RAM, which introduces substantial additional latency during read operations, thereby contributing to higher compute latency. Specifically, when recomputation occurs, the weights must be refetched from flash memory, which could disrupt cache hits and lead to increased overall latency. Despite this discrepancy, our method still generates fusion

*Table 2.* The different microcontrollers & boards used in our experiments.

| Board | MCU | Core | RAM (kB) | Flash (kB) |
|---|---|---|---|---|
| Nucleo-f767zi | STM32F767ZI | Cortex-M7 @ 216 MHz | 512 | 2048 |
| Stm32f746g-disco | STM32F746NG | Cortex-M7 @ 216 MHz | 320 | 1024 |
| Nucleo-f412zg | STM32F412ZG | Cortex-M4 @ 100 MHz | 256 | 1024 |
| esp32s3-devkit | ESP32-S3-WROOM-1N8 | Xtensa @ 240 MHz | 512 | 8192 |
| esp32c3-devkit | ESP32c3-1-MINI-M4N4 | RISC-V @ 160 MHz | 384 | 4096 |
| hifive1b | SiFive FE310-G002 | RISC-V @ 320 MHz | 16 | 4096 |

*Table 3.* Minimal peak RAM use, measured in kB.
(Vanilla: un-fused model)

| | MBV2-w0.35 | MN2-vww5 | MN2-320K |
|---|---|---|---|
| *(Fusion)* | | | |
| Vanilla | 194.44 | 96 | 309.76 |
| MCUNetv2 | 63 | 45 | 215 |
| StreamNet | 66 | 44 | 208 |
| **msf-CNN** | 8.56 | 15.368 | 51.164 |

*Table 4.* Inference execution time, measured in *ms*, with msf-CNN tuned with minimal peak RAM. (OOM: Out-of-Memory)

| | MBV2-w0.35 | MN2-vww5 | MN2-320K |
|---|---|---|---|
| *(MCU)* | | | |
| stm32f767 | 1996.8 | 1723.0 | 19329.9 |
| stm32f746 | 1379.6 | 1727.5 | 16261.9 |
| stm32f412 | 5270.1 | 4943.4 | 56979.0 |
| esp32s3 | 6748.2 | 5974.1 | 76763.6 |
| esp32c3 | 6792.7 | 6248.9 | 73713.8 |
| SiFive | 10000.0 | OOM | OOM |

configurations for the MBV2-w0.35 and MN2-vww5 models that outperform MCUNetV2. Particularly for memory-sensitive but time-insensitive applications, we can set the constraint $F_{max}$ to infinity, thereby obtaining novel fusion configurations with minimal RAM usage.

## 9. Discussion

Our experiments demonstrate msf-CNN's capability to optimize resource usage with diverse CNN models, under user-specified constraints emphasizing either compute latency or RAM footprint. Furthermore, msf-CNN generates code that is deployable across diverse microcontroller ISAs. Users can thus produce optimal CNN fusion configurations tailored to specific industrial hardware requirements. However, some limitations remain, on which our future work will focus next.

**Parameter Space –** The current optimization scope is limited to fusion block positioning and depth selection, with the number of output elements per iteration fixed at one. This parameter significantly impacts both memory footprint

and compute overhead, which warrants further exploration.

**Caching Paradigm –** The search space currently incorporates only the H-cache paradigm. Future implementations should integrate alternative caching strategies to enhance optimization flexibility.

**Neural Network Architecture –** The work currently focuses exclusively on convolutional neural network architectures (CNNs). The analysis of other prevalent structures, particularly attention mechanisms and recurrent neural networks (RNNs), remains an open research direction.

## 10. Related Work

**Machine Learning Compilers for MCUs –** Compilers such as Tensor Virtual Machine (TVM)(Chen et al., 2018), IREE(The IREE Authors, 2019), FlexTensor (Zheng et al., 2020), and Buddy (Zhang et al., 2023) offer automated transpilation and compilation for models produced by major Machine Learning (ML) frameworks, including TensorFlow and PyTorch. As an extension of TVM, microTVM provides low-level optimizations and routines tailored for execution on various processing units, including a wide range of microcontrollers. Other prior work such as RIOT-ML (Huang et al., 2024) combine a small general-purpose OS with microTVM, for comprehensive support for ML frameworks and operator implementation on divers MCUs. Similarly, msf-CNN work utilizes microTVM as both the front-end importer for model files and the code generator for low-end platforms. However, none of the above tools provide CNN fusion optimization mechanisms, in contrast to msf-CNN.

**Efficient Neural Network Structure –** For models to operate on low-power IoT devices, they must be compact and computationally efficient. Studies have demonstrated the use of lightweight CNNs for speech recognition and age classification (Maayah et al., 2023), water leakage detection (Atanane et al., 2023), fall detection for the elderly (Fang et al., 2021) and other tasks (Hussain & Haque, 2018; Zhu-Zhou et al., 2023). Tiny vision transformers have also been employed for classification tasks in various studies (Jinyang Yu et al., 2023; Liang et al., 2023; Yao & Liu, 2023; Wyatt et al., 2021). Besides handcrafting a lightweight structure by reducing layer number or kernel size, people (Iandola

*Table 5.* Optimal fusion settings on Nucleo-f767zi. RAM (kB), Latency (ms). SAA: Same as above. **Bold: msf-CNN beats MCUNetv2**.

| | | MBV2-w0.35 | | MN2-vww5 | | MN2-320K | |
|---|---|---|---|---|---|---|---|
| | | RAM | Latency | RAM | Latency | RAM | Latency |
| Vanilla | | 194.44 | 807.6 | 96 | 509.7 | 309.76 | 4394.3 |
| MCUNetv2 | | 63 | 1513.0 | 45 | 810.0 | 215 | 2777.0 |
| StreamNet | | 66 | 417.0 | 44 | 225.0 | 208 | 1444.0 |
| P1: Min. RAM s.t. Compute Cost Limit | | | | | | | |
| $F_{max}$ | 1.1 | 67.996 | 961.9 | **45.283** | **696.0** | 199.6 | 4171.0 |
| | 1.2 | (SAA) | | **26.24** | **769.2** | 196.072 | 4525.1 |
| | 1.3 | **21.389** | **1313.8** | 20.568 | 922.7 | 195.333 | 4680.7 |
| | 1.4 | **15.199** | **1412.3** | 17.904 | 931.3 | 156.864 | 5128.9 |
| | 1.5 | (SAA) | | (SAA) | | 94.224 | 5370.3 |
| | Inf | 8.56 | 1996.8 | 15.368 | 1723.0 | 51.164 | 19329.9 |
| P2: Min. Compute Cost s.t. RAM Limit | | | | | | | |
| $P_{max}$ | 16 kB | 15.199 | 1412.3 | 17.904 | 931.3 | (No Solution) | |
| | 32 kB | **25.803** | **1266.3** | **26.24** | **769.2** | | |
| | 64 kB | **63.603** | **1121.7** | **45.283** | **684.6** | 63.456 | 9458.6 |
| | 128 kB | 83.133 | 947.0 | 89.6 | 683.4 | 94.224 | 5370.3 |
| | 256 kB | 181.44 | 879.2 | (SAA) | | 247.808 | 3923.2 |

et al.; Tan & Le; Howard et al., b; Sandler et al.; Howard et al., a) also re-designed the basic blocks to replace common convolutions for lower memory footprint and compute latency.

**Tiny Neural Architecture Search (NAS)** This technique is employed to automatically search for model structures with optimal accuracy under the constraints of memory, flash footprint and compute latency. TinyNAS (Lin et al., 2020) and the Once-for-All Network (Cai et al., 2019) leverage Neural Architecture Search (NAS) to design CNNs with exceptionally small memory requirements for MCUs. The resulting networks require only a few hundred kilobytes of RAM for execution. However, contrary to msf-CNN, these methods necessitate retraining or fine-tuning of pre-existing networks.

**Memory Optimization for CNN layers –** Memory optimization strategies can be broadly categorized into scheduling-based and fusion-based methods. Scheduling-based methods, such as those implemented in frameworks like vMCU (Zheng et al., 2024b), MoDEL (Steiner et al., 2023) and TinyEngine (Lin et al., 2021), focus on the efficient reuse of memory pools to minimize peak memory usage by leveraging the different lifetimes of inter- and intra-layer tensors. For instance, TinyEngine employs in-place tensor updates for depthwise convolutions, enabling the corresponding input and output tensors to share the same memory space. vMCU further generalized this approach for common convolution and pooling operations. Although both methods achieve a peak memory reduction exceeding 50%, they still generate a complete output tensor for each layer. This requirement remains problematic for low-power MCUs with limited RAM, particularly when dealing with large input sizes or an extensive number of output channels. Prior work on fusion was covered in Section 2. Contrary to msf-CNN, these methods do not fully exploit the potential of multiple fusion blocks.

## 11. Conclusion

In order to fulfill the full potential of AI, convolutional neural networks (CNNs) must not only execute in the cloud or on edge computing gateways, but also on the smaller microcontroller-based devices which take part in our cyber-physical systems. These small energy-efficient devices pose a great challenge regarding the joint optimization of RAM memory consumption and inference latency for CNNs. In this context, we presented msf-CNN, a technique and heuristics able to identify pools of practical fusion-based optimizations for CNN inference which jointly satisfy memory and latency constraints. Compared to previous work on CNN fusion for microcontrollers, msf-CNN identifies a wider set of applicable solutions, on more diverse hardware. Our experimental evaluation using the open source implementation we provide for common microcontrollers (ARM Cortex-M, RISC-V, and ESP32) show that msf-CNN can achieve inference with less than 50% the peak RAM usage state-of-the-art. As such msf-CNN provides a new level of flexibility for embedded system designers, which can now better tune the trade-off between peak RAM and model inference latency on various MCUs.

## Impact Statement

This paper presents work contributing to the field of Machine Learning on small microcontrollers. There are many potential societal consequences of our work, none which we feel must be specifically highlighted here.

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

**Code Availability –** The implementation of msf-CNN is publicly available. Please check the supplementary material for the URL.

# A. Analysis of the number of MAC operations

Analyzing the number of MAC operations in the fusion block is quite complex. The input tensor for each layer is sliced into overlapped tiles, and the kernel performs convolution on the data within each tile. Here, the number of overlapped tiles $N^{tile}$ of each layer is

$$N^{tile} = \lfloor \frac{h^{in} + 2p - t}{s^{tile}} + 1 \rfloor \lfloor \frac{w^{in} + 2p - k}{s^{layer}} + 1 \rfloor, \quad (14)$$

where $h^{in}, w^{in}$ are the height and width of input tensor, $s^{tile}, s^{layer}$ are the stride of tile and layer, $p$ represents the input padding. Recall that $t, k$ are the tile size and kernel size respectively.

And the output size of each tile is determined as:

$$O^{tile} = \lfloor \frac{t - k}{s^{layer}} + 1 \rfloor c^{out}. \quad (15)$$

whereby $c^{out}$ is the number of output channels. We can therefore derive the number $C^{layer}$ of MAC operations of a fused convolutional layer as:

$$C^{layer} = N^{tile} \times O^{tile} \times k^2 \times c^{out}. \quad (16)$$

Finally, we can derive $C^{fb}$ the total MAC operations of the entire fusion block as:

$$C^{fb} = \sum C^{layer}. \quad (17)$$

