# OpenReview forum: "msf-CNN: Multi-Stage Fusion with  Convolutional Neural Networks for TinyML"
_ICML.cc/2025/Conference — Submitted to ICML 2025_

### Official Review · Reviewer_xtNR · 2025-03-16

**Overall Recommendation:** 3

**Summary:**

The paper presents a multi-stage fusion technique for optimizing CNN inference on memory-constrained microcontrollers (MCUs), called msf-CNN. The main objective of this work is to efficiently execute deep neural networks on resource-limited IoT devices by reducing RAM usage through layer fusion while balancing inference latency. The proposed method formulates the problem as a shortest path search in a directed acyclic graph (DAG) to identify optimal fusion configurations. The proposed method is validated on various MCU platforms (ARM Cortex-M, RISC-V, ESP32), showing significant reductions in RAM usage with up to 50% less than existing methods like MCUNetV2 and StreamNet.

**Claims And Evidence:**

Overall, the paper backs up most of its claims with solid evidence, but a few could be clearer. The claim that msf-CNN reduces RAM usage by up to 50% is mostly supported, but the actual savings vary by model, sometimes being much lower. The idea that msf-CNN introduces a new trade-off between memory and compute overhead is valid, but the impact on real-time performance isn't fully explored, especially since inference latency can increase in extreme cases.

**Essential References Not Discussed:**

No

**Experimental Designs Or Analyses:**

Yes, they all make sense.

**Methods And Evaluation Criteria:**

Yes, for the most part. The main objective of the evaluation in this work is to explore different trade-offs between RAM usage and compute overhead through a thorough analysis across multiple MCU architectures (ARM Cortex-M, RISC-V, ESP32). The comparison against baselines like MCUNetV2 and StreamNet is well-structured, and the use of RAM and compute latency measurements is appropriate for this type of application. However, the efficiency of the proposed method could be clearer if the trade-off between RAM and compute latency were visualized as a curve, ideally with a Pareto-optimal front to better illustrate the best achievable balance between memory and performance. This would provide a more intuitive understanding of how msf-CNN performs under different constraints and would make it easier to compare against prior works.

**Other Comments Or Suggestions:**

N/A

**Other Strengths And Weaknesses:**

Strengths:


-- The use of graph-based optimization to find efficient fusion settings is novel.


-- There is a significant decrease in peak RAM usage (65-87%), making CNN inference feasible on extremely memory-constrained MCUs.


-- The paper provides mathematical formulations for peak RAM and compute cost.


Weaknesses:


-- The current formulation only applies to CNNs and does not extend to other architectures like transformers, RNNs, or hybrid models.


-- The trade-off between latency and memory could have been better presented as a curve rather than a table.


-- While the shortest path problem is well-explained, the complexity analysis of the search algorithm is not fully detailed.


-- Energy consumption is another important factor for edge devices. An analysis of energy consumption under different fusion configurations would strengthen the practical implications of the work.

**Questions For Authors:**

See my points listed as weaknesses.

**Relation To Broader Scientific Literature:**

This paper builds on prior works including MCUNetV2 and StreamNet by enabling deeper fusion blocks and using a graph-based shortest-path approach instead of brute-force searches. It formulates fusion as a DAG-based optimization problem, improving RAM efficiency while balancing compute overhead. Unlike Neural Architecture Search, which redesigns models, msf-CNN optimizes memory allocation for existing CNNs. It also introduces iterative pooling and dense layers, reducing memory use similar to streaming CNN architectures. While msf-CNN improves memory efficiency, energy consumption analysis and support for non-CNN models could further connect it to real-world TinyML applications.

**Theoretical Claims:**

No, I didn't thoroughly checked the correctness of theoretical analysis in this paper.

---

> ### Author Rebuttal · Authors · 2025-04-01
>
> Thank you very much for the insightful feedback on our manuscript. Below, we address each of your comments and questions in detail.
>
> **[C1. The current formulation only applies to CNNs and does not extend to other architectures like transformers, RNNs, or hybrid models.]**
>
> Thanks for the suggestion. We acknowledge that our current work focuses exclusively on CNNs. As stated in Section 9, we are actively working to extend the methodology to other architectures (e.g., transformers, RNNs).
>
> **[C2. The trade-off between latency and memory could have been better presented as a curve rather than a table.]**
>
> Thanks for the suggestion. Here we visualized the RAM-latency trade-off in [anonymous link](https://anonymous.4open.science/r/msf-CNN-3BE5/RAM-latency%20trade-off.pdf).
>
> **[C3. While the shortest path problem is well-explained, the complexity analysis of the search algorithm is not fully detailed.]**
>
> Thanks for the remark. We will add a more detailed analysis on computational complexity in the appendix.
>
> We provide below a quick preliminary analysis of the *worst-case scenario*. We also highlight that these shortest path computations do not take place on the microcontroller at runtime, but offline on a PC (which expands the realm of what can be assessed as bearable computation).
>
> First, we consider the *lower-bound* of the search algorithm. As shown in Section 6, both problem P1 and P2 without constraints can be transformed into a multiple single-source-single-target shortest path problem, which can by solved by Dijkstra's algorithm with Fibonacci heap [1] with complexity:
>
> $$
> O(E + V log(V)),
> $$
>
> where $E$ and $V$ denotes edges (possible fusion blocks) and vertices (layers) of the DAG. In the worst case $E=\sum_{n=1}^V (n-1)$, which leading the *overall lower-bound to $O(V^2)$*.
>
> Concerning Problem P1 with constraints: if we don't prune the search space (iteratively), we need to brute force all possible fusion settings of the DAG to form a subspace that fulfills the latency constraints. This leads to enumerating all simple paths from input layer to the output layer. In the worst-case, the $i$-th layer of an $N$-layered CNN has $2^{i-2}$ paths from input layer pointing to it. Thus, starting from the input layer, we obtain $2^{N-2}$ fusion combinations, which becomes unbearable for a deep neural network.
>
> Hence, we apply a pruning strategy (see Equ. 11-13, line 177-183) to reduce the complexity from $O(2^{N-2})$ to $O(N^2)$. The idea: we erase the edges with maximal RAM usage per iteration. In the worst-case, only one edge is erased in each iteration, with a complete DAG with $\frac{N(N-1)}{2}$ edges. Thus, the worst-case complexity of our search algorithm for constrained Problem P1 is $O(N^2)$.
>
>
> **[C4. An analysis of energy consumption under different fusion configurations would strengthen the practical implications of the work.]**
>
> While energy-efficiency is important as pointed out, our work primarily targets RAM budget constraints rather than energy budgets. Fitting model operation within the limits of a tiny maximal memory budget is the critical first hurdle faced by deep edge AI developers. Our approach is particularly relevant for small devices where RAM is the first-order bottleneck (e.g., MEMS and tiny sensors).
>
> We will tackle energy consumption measurements in future work, as suggested. We acknowledge that energy consumption is highly hardware- and use-case-dependent, as well as influenced by computation latency and the amount of RAM used.
>
> For completeness, we will add a short discussion in the appendix on this aspect, without blurring the main purpose of our work.
>
> Thank you again for your valuable feedback. Please let us know if you have further questions.
>
> **Reference**
>
> [1] Fredman, Michael L., and Robert Endre Tarjan. "Fibonacci heaps and their uses in improved network optimization algorithms." Journal of the ACM (JACM) 34.3 (1987): 596-615.

---

### Official Review · Reviewer_v2SP · 2025-03-17

**Overall Recommendation:** 3

**Summary:**

In this paper, the authors introduce msf-CNN, a multi-stage fusion (msf) approach that identifies optimal fusion settings for CNNs by navigating the fusion solution space represented as a directed acyclic graph (DAG). The goal is to reduce RAM usage without introducing significant computational overhead. The msf-CNN was evaluated on various MCU platforms, including ARM Cortex-M, RISC-V, and ESP32, and demonstrated up to a 50% reduction in RAM usage compared to previous approaches such as MCUNetV2 and StreamNet.

**Claims And Evidence:**

Many claims made in the paper are supported by clear and convincing evidence, except:

1. It claims that msf-CNN can generalize beyond CNNs, but no experiments on non-CNN architectures are provided.

2. It fails to clarify the trade-offs between RAM reduction and inference latency. For instance, Table 4 shows a consistent 2-5x increase in inference latency, which contradicts the central claim that msf-CNN simultaneously achieves both low memory and low latency.

**Essential References Not Discussed:**

The paper “μNAS: Constrained Neural Architecture Search for Microcontrollers” [arXiv:2010.14246] was published in the Proceedings of the 1st Workshop on Machine Learning and Systems in 2020. In this paper, neural networks are represented as DAGs, with nodes and edges representing operators (layers) and their connectivity.

Please mention this paper and give them appropriate credits.

**Experimental Designs Or Analyses:**

Their main experimental designs or analyses align well with the goal of optimizing CNNs for TinyML deployment. But let’s reiterate the insufficiencies:

1. The paper claims that msf-CNN can generalize beyond CNNs, but no experiments on non-CNN architectures are provided.

2. The paper fails to clarify the trade-offs between RAM reduction and inference latency. For instance, Table 4 shows a consistent 2-5x increase in inference latency, which contradicts the central claim that msf-CNN simultaneously achieves both low memory and low latency.

3. The paper only evaluates on MobileNetV2 and MCUNet, which are popular models, but there is no evaluation on standard TinyML benchmark datasets.

4. Significant reduction in RAM usage is undoubtedly important, but without showing accuracy changes at the same time, their study is incomplete.

**Methods And Evaluation Criteria:**

Convincing:

1. Under the DAG representation, the fusion block optimization problem is formulated as the  shortest-path problem. In addition, the pruning strategy reduces the search complexity from O(2^N) to O(N^2). These techniques are both novel and practical for TinyML deployment.

2. Their focus on peak RAM usage (rather than average RAM usage) is meaningful for avoiding memory overflow on MCUs, which is often the bottleneck in edge devices.


Not convincing:

1. The paper only evaluates on MobileNetV2 and MCUNet, which are popular models, but there is no evaluation on standard TinyML benchmark datasets.

2.  Significant reduction in RAM usage is undoubtedly important, but without showing accuracy changes at the same time, their study is incomplete.

**Other Comments Or Suggestions:**

None

**Other Strengths And Weaknesses:**

None

**Questions For Authors:**

Why is model accuracy not reported in the experimental results?

**Relation To Broader Scientific Literature:**

Prior work, such as MCUNetV2 (NeurIPS 2021) and StreamNet (NeurIPS 2024), introduced basic layer fusion for reducing RAM usage on MCUs.

What msf-CNN excels is that it extends previous effort by performing multi-stage fusion and formulating the fusion process as the shortest-path problem on DAG. This approach makes the whole problem more efficient and scalable, leading to 50% more RAM reduction.

**Theoretical Claims:**

It makes sense for the authors to formulate the fusion block optimization problem as the shortest-path problem under the DAG representation. It is also true that the pruning strategy reduces the search complexity from O(2^N) to O(N^2). While this may not be highly problematic, the pruning strategy is not guaranteed to find the global optimal fusion setting though.

Their analyses on RAM usages in “Iterative Computation of Global Pooling” and “Iterative Computation of Dense Layer” are valid. But the analysis on the corresponding computational cost is missing and thereby unconvincing.

---

> ### Author Rebuttal · Authors · 2025-04-01
>
> Thank you very much for the insightful feedback on our manuscript. Below, we address each of your comments and questions in detail.
>
> **[Q1. Why is model accuracy not reported in the experimental results?]**
>
> We appreciate the reviewer raising this, as it allows us to clarify a crucial aspect of our method. msf-CNN is a **computation scheduling and memory optimization technique that does _not_ alter the model's architecture, parameters, or the mathematical operations performed.** It only changes _when_ and _how_ intermediate results are computed and stored to minimize peak memory. Therefore, the final output, and consequently the model's accuracy, remains **identical** to the original, unfused model. Hence, standard ML performance benchmarks focusing on accuracy are irrelevant here.
>
> Accordingly, we will add disambiguating text in the paper, explicitly mentioning the above.
>
> **[C1. Missing Reference to μNAS]**
>
> We thank the reviewers for pointing out this paper. We agree that μNAS is a seminal work in representing neural networks as directed acyclic graphs (DAGs) and has inspired many subsequent studies, including our own. We will add a citation to μNAS in the related work section and acknowledge its contributions.
>
> Although both works focus on reduction of peak memory usage and ultilized DAG to model the problem, μNAS addresses it by altering the neural network architecture and reordering the execution of operators, while msf-CNN developed a graph-based solver to find fusion settings that have optimal RAM-latency trade-off under specific constratins. We believe both methods are orthogonal and can be applied simultaneously.
>
> **[C2. The paper claims that msf-CNN can generalize beyond CNNs, but no experiments on non-CNN architectures are provided.]**
>
> We would like to clarify that this work currently focuses exclusively on convolutional neural network architectures (CNNs). Nevertheless, as stated in the future work we described (in Section 9) we are currently working on generalizing our framework to other architectures.
>
> **[C3. The paper fails to clarify the trade-offs between RAM reduction and inference latency...which contradicts the central claim that msf-CNN simultaneously achieves both low memory and low latency.]**
>
> We noticed the ambiguity in our initial phrasing. We acknowledge that multi-stage fusion inherently involves a trade-off: reducing peak RAM often requires recomputation, which increases inference latency.
>
> Our central claim is _not_ that msf-CNN simultaneously achieves the low memory and low latency compared to all baselines in all scenarios. Instead, msf-CNN provides a **systematic framework (the DAG-based search) to explore the RAM-Latency Pareto frontier** more effectively than prior methods. This allows practitioners to find _optimal trade-offs_ that meet specific constraints. More details can be found in Table 5.
>
> **[C4. The paper only evaluates on MobileNetV2 and MCUNet, which are popular models, but there is no evaluation on standard TinyML benchmark datasets.]**
>
> Please check the above response to Q1. Accuracy is systematically unchanged, hence standard accuracy-focused benchmark suites are irrelevant for our purpose.
>
> **[C5. Significant reduction in RAM usage is undoubtedly important, but without showing accuracy changes at the same time, their study is incomplete.]**
>
> Please check the above response to Q1. Accuracy remains strictly unchanged, hence additional accuracy measurements are not needed.
>
> **[C6. Missing Computational Cost Analysis for Iterative Computations]**
>
> Figures 3 and 4 illustrate our technique for global pooling and dense layers, wherein computation is partitioned into several iterations synchronized with partial outputs from the preceding fusion block. By processing input vectors element-by-element, this approach calculates the output iteratively while preserving the exact set of arithmetic operations. Therefore, the total number of MACs is identical to the baseline, resulting in zero computational overhead.
>
>
> Thank you again for your valuable feedback. Please let us know if you have further questions.

---

> > ### Comment · Reviewer_v2SP · 2025-04-04
> >
> > The authors have clarified most of my concerns to some extent, and I would now recommend a "weak accept".

---

> > > ### Author Response · Authors · 2025-04-09
> > >
> > > Thank you very much for your feedback and for acknowledging our clarifications in the rebuttal. We sincerely appreciate your updated "weak accept" recommendation. However, we notice the system still displays the original "weak reject" score. Could there be a technical issue preventing the update, or do we need to take any specific action? Please forgive our potentially naive question on this.

---

### Official Review · Reviewer_RM2Q · 2025-03-17

**Overall Recommendation:** 4

**Summary:**

msf-CNN is a framework for reducing CNN memory usage on very small devices (e.g., MCUs) through multi-stage layer fusion. The authors represent CNN layers as edges in a directed acyclic graph, then systematically search for fusion “blocks” using graph-based algorithms to minimize peak RAM or computation cost. They also introduce iterative global pooling and dense layers to further reduce memory. Experiments on multiple MCU architectures (Cortex-M, RISC-V, and ESP32) show that msf-CNN cuts RAM usage by 50% or more compared to leading approaches, at the price of potentially higher latency. The system lets designers choose whether to favor minimal memory or minimal computational overhead, broadening CNN deployment possibilities for highly constrained hardware.

Update after rebuttal: Thanks authors for the replies which resolve my concerns, my original recommendation decision remain the same.

**Claims And Evidence:**

There are 3 major claims (6 in total mentioned in paper)

Claim 1: msf-CNN yields substantial RAM savings over existing solutions.
Evidence for this appears in Section 8 (Experiments on Microcontrollers), where the authors compare msf-CNN to MCUNetV2 and StreamNet. For example, Table 3 shows that msf-CNN can achieve as little as 8.56 kB of RAM usage for a MobileNetV2 variant that un-fused baselines (and even state-of-the-art fusion approaches) cannot shrink below 60–65 kB. These are measured results on real boards (esp32s3, STM32, RISC-V boards).

Claim 2: msf-CNN allows flexible trade-offs between memory and computation cost.
Evidence is given in the tables of Section 6.3 and expanded in Section 8, where setting different constraints for peak RAM usage or latency leads to different—but valid—fusion strategies. Under one scenario, msf-CNN can reduce memory usage at the cost of 2× to 5× extra computations; in another, it can optimize for minimal overhead by allowing a higher memory budget. Experimental data confirm that even with a strict memory limit, msf-CNN still finds feasible solutions.

Claim 3: msf-CNN is portable to a broad range of microcontrollers.
Section 8 illustrates results on MCUs from various families (ARM Cortex-M7, Cortex-M4, ESP32, RISC-V), showing that the same approach can be applied without major changes in code generation. The authors also cite in Section 7 (Implementation Details) that they rely on the microTVM code generator, then custom rewrite the fusion blocks.

Overall, the paper’s core claims are well supported by experimental evidence and fairly thorough memory and compute measurements.

**Essential References Not Discussed:**

One relevant area for further contextualization is dynamic scheduling approaches (e.g., memory re-use optimization beyond standard layer-by-layer tiling) in compilers like Apache TVM’s “AutoScheduler” or advanced memory planning from specialized frameworks.

**Experimental Designs Or Analyses:**

The paper’s experiments (Section 8) involve:
1. Deploying three distinct CNN backbones (MobileNetV2 variants and MCUNetV2) with different image sizes and channel scaling.
2. Measuring memory usage and latency on real MCU hardware across ARM, RISC-V, and Xtensa.
3. Comparing msf-CNN with two prior methods: MCUNetV2 and StreamNet.
These experiments convincingly show the memory savings as well as the performance overhead. The authors also explore different constraints (peak memory limit or overhead limit) to demonstrate msf-CNN’s flexibility. The discussion of observed latency variations across architectures and clock speeds shows thoroughness. The design is sound, covering multiple MCU families and providing repeated references to measured rather than purely simulated results, which is a strong point.
A minor improvement in the analysis might be to include details about standard deviation of runtime or memory measurements. But overall, the design is robust, and the analyses in Section 8 reasonably validate the authors’ claims.

**Methods And Evaluation Criteria:**

The authors base their methodology on re-creating an inverted dataflow graph of the CNN (Section 5), encoding possible single-layer and multi-layer fused operations as edges. They then use a “shortest path” perspective to find the minimal peak memory route or the minimal total computation route through the graph. The authors’ evaluation criteria focus primarily on:
1. Peak RAM usage (with direct measurement of actual memory consumption at runtime).
2. Computation overhead (expressed in MAC counts as well as real-world latency).
3. Portability (testing on different MCUs with different memory and CPU constraints).
This set of evaluation criteria is quite suitable for the problem at hand: microcontroller-based deployment demands a careful balance of memory versus latency. The authors also clearly state that they are not primarily targeting top classification accuracy or novel neural architectures—rather, they concentrate on making standard CNNs feasible on extremely constrained hardware.

**Other Comments Or Suggestions:**

1. Clarify memory measurement procedures: It would be helpful to describe how precisely the peak memory usage is measured at runtime—some extra detail on whether instrumentation or static analysis was used could increase reproducibility.
2. Highlight impact of flash memory reads: The authors note (Section 8.3) that re-fetching weights from flash can degrade performance. A small dedicated subsection about practical flash-read overhead on MCUs would be valuable, especially for new TinyML practitioners.

**Other Strengths And Weaknesses:**

NA, already pretty much been covered by previous questions.

**Questions For Authors:**

1. Has msf-CNN been tested on advanced caching paradigms, beyond H-cache?
If so, do you anticipate further memory savings or latency improvements by partial caching along both dimensions (height and width) at different layers?
Potential Effect on Evaluation: This would show how future expansions might yield intermediate solutions with moderate memory usage and moderate overhead.

2. Could msf-CNN fuse multiple non-convolutional layers?
Many CNN-based networks now incorporate activation layers or attention blocks. Is there a straightforward extension of your DAG approach to these operators?
Potential Effect on Evaluation: It may confirm that the approach is robust beyond typical CNN layer sequences.

**Relation To Broader Scientific Literature:**

The paper builds on a line of work dealing with memory minimization and scheduling for neural networks on FPGAs and GPUs, referencing Alwani et al. (2016) for fused-layer approaches and subsequent expansions, as well as past MCU-specific optimization frameworks like MCUNetV2 and StreamNet. It also integrates with the general domain of “TinyML compilers” such as microTVM and IREE, stating clearly that existing compilers lacked multi-block fusion optimization.
Compared to prior studies, msf-CNN adds:
1. A more comprehensive search space (rather than a single big fuse block or a forced partial fusion).
2. A graph-based solver for user-driven constraints (peak memory or time).
3. Extensions to iterative global pooling and dense layers to reduce memory usage further.

**Theoretical Claims:**

The theoretical foundation rests upon:
1. Graph interpretation of CNN layers (Section 5): The authors describe each layer or fused block as an edge with associated “peak RAM usage” and “MAC cost.”
2. Shortest-path or minimax path approach (Sections 5 and 6): For instance, minimizing peak memory is transformed into a minimax path problem, solvable with minor modifications to standard algorithms like Dijkstra’s or BFS/DFS-based searches.
These approaches are correct at a conceptual level, and the paper does not contain advanced new proofs.

---

> ### Author Rebuttal · Authors · 2025-04-01
>
> Thank you very much for the insightful comments and constructive suggestions on our work. Below, we address each of your comments and questions in detail.
>
> **[C1. Essential References Not Discussed]**
>
> Sorry for the omitting the mention of TVM's "AutoScheduler". We will definitely mention it in the paper's final version.
>
>
> **[C2. Clarify memory measurement procedures]**
>
>  We relied on TVM's [Ahead-of-Time (AoT)](https://discuss.tvm.apache.org/t/implementing-aot-in-tvm/9206) compilation for model code generation, and enabled [Unified Static Memory Planning (USMP)](https://discuss.tvm.apache.org/t/rfc-unified-static-memory-planning/10099) for static memory allocation. So we can fetch the runtime RAM usage directly from that memory planner. We will add this details into the updated manuscript.
>
> **[C3. Highlight impact of flash memory reads]**
>
> Thanks for pointing this out. Currently we are also conducting experiments to more precisely measure this effect. We will add a short discussion on this in our manuscript.
>
> **[Q1. Has msf-CNN been tested on advanced caching paradigms, beyond H-cache? ]**
>
> Yes. Our solver is designed to easily extend to alternative caching schemes and varying input patch sizes. Furthermore, we are actively exploring a **mixed-mode caching strategy** within individual fusion blocks (e.g., employing distinct caching schemes across sequential layers). So the search space is enlarged to yield a better RAM-Latency trade-off. We will definitely add corresponding experimental results in appendix of our manuscript, if initial findings are validated before the deadline.
>
> **[Q2. Could msf-CNN fuse multiple non-convolutional layers?]**
>
> Yes. For activation layers, our method is orthogonal to kernel fusion and they can be applied concurrently. For example, a convolutional layer followed by Batch Normalization and ReLU will be first merged into one operator (kernel fusion, Conv+BN+ReLU), and try to fuse with other convolutional layers by our msf-CNN. We just need to add latency and RAM usage estimator of the activation layers onto our DAG-based solver.
>
> However, we acknowledge that currently our method only discuss the fusion of CNN-based layers.  We will further explore the fusion potential of non-CNN architectures like RNN, attention blocks etc.
>
> Thank you again for your valuable feedback an suggestions. Please let us know if you have further questions.

---

### Official Review · Reviewer_UtUu · 2025-03-18

**Overall Recommendation:** 2

**Summary:**

This work proposes a multi-stage fusion method, called MSF-CNN, to optimize RAM usage for tiny CNNs on microcontrollers. By modeling fusion as a graph optimization problem, it minimizes memory and computation costs. Experiments on various MCUs demonstrate that MSF-CNN significantly reduces peak RAM usage, providing flexible trade-offs for real-world TinyML applications.

**Claims And Evidence:**

The claims are clear and supported by theoretical/empirical analysis.

**Essential References Not Discussed:**

N/A.

**Experimental Designs Or Analyses:**

Yes. This work analyzes the peak memory usage and latency of different fusion strategies.

**Methods And Evaluation Criteria:**

Yes, the evaluation criteria generally makes sense.

**Other Comments Or Suggestions:**

It would be helpful to provide background information on the implementation of fusion.

**Other Strengths And Weaknesses:**

**Strengths:**

1. The proposed framework is generally interesting and sound, with the potential to serve as a tool for determining the optimal trade-off between memory and latency when using fusion.

**Weaknesses:**

1. The major concern is whether the proposed method can practically improve latency under a given peak memory constraint compared to previous heuristic solutions. For example, MCUNet identifies that peak memory usage occurs in the first layers, a common pattern in most CNNs, and employs a simple fusion strategy to reduce it. I suspect such a heuristic solution could be applicable to most CNNs, making the proposed framework less practically useful. The authors should demonstrate whether the search space is large and whether a heuristic or simple strategy performs sufficiently well, which is not addressed in the current paper.

2. This work lacks sufficient benchmarking: no comparison has been provided regarding latency relative to previous methods. For example, only peak memory usage is compared with MCUNet, whereas ultimately, overall latency under the peak memory constraint is what truly matters.

3. The proposed solution is generally intuitive, and this paper would be strengthened if the authors leveraged the framework to analyze the fusion search space.

**Questions For Authors:**

My questions have been included in the weakness section.

**Relation To Broader Scientific Literature:**

This work can serve as a tool for deploying tiny CNNs, helping to determine the optimal trade-off between memory and latency when using fusion.

**Theoretical Claims:**

N/A

---

> ### Author Rebuttal · Authors · 2025-04-01
>
> Thank you very much for the insightful remarks on our manuscript. We address below your questions/comments:
>
> **[C1. Can msf-CNN practically improve latency under a given peak memory constraint compared to previous heuristic solutions? ]**
>
> We agree that this is a key consideration.
>
> Compared to prior work using simpler heuristic strategies (e.g. MCUNet) we wish to highlight the following:
>
> - **Experimental Evidence (Table 5):** our measurements show that msf-CNN **can find solutions with much lower peak memory usage** compared to MCUNet (though sometimes potentially at the cost of higher computational latency, illustrating a different trade-off point). This indicates that even for seemingly simple fusion problems, a systematic search can yield better memory optimization than a simplified heuristic, justifying the utility of our approach.
>
> - **Enlarged Search:** For an N-layer network (excluding input/output layers for fusion), there exist $2^{N-2}$ potential fusion configurations. When facing **extremely tight resource constraints** (e.g., MCUs with RAM < 50 kB), heuristics considering only a small subset of configurations e.g. the initial few layers as in MCUNet, are often **insufficient** to meet the memory budget. In such stringent scenarios, exploring more complex, multi-stage fusion strategies becomes essential, which is precisely where msf-CNN excels by providing a structured method to navigate this larger space.
>
> - **New Tradeoffs:** In some cases, a latency slump may be tolerable if memory is significantly reduced, within some critical thresholds. A concrete example: the audio signal analysis use-case described in [1]. In such use-cases, provided inference still completes within "real-time" execution bounds, a trade-off could be considered a good deal on a small microcontroller if the reduced memory footprint now fits within the total available RAM budget -- whereas previously it did not -- even if inference is slower
>
> **[C2. Benchmarks lack latency measurements comparing to prior work.]**
>
> **Table 5** provides measurement of latency with msf-CNN compared to prior work (MCUNetV2) under various constraint settings. **This table explicitly lists both peak memory usage and the corresponding inference latency for different configurations**. More specifically, the results presented in Table 5 (**marked in bold**) clearly show how, under certain constraints, the fusion configurations identified by msf-CNN achieve both lower latency and RAM usage compared to MCUNetV2.
>
> **[C3. The proposed solution is generally intuitive, and this paper would be strengthened if the authors leveraged the framework to analyze the fusion search space.]**
>
> Thank you for the suggestion, which can indeed be the subject for future work. In this paper, we have focused on the primary challenge, i.e., systematically determining the _optimal_ multi-stage fusion strategy, especially when considering the trade-off between peak memory, computational latency, and specific hardware characteristics.
>
> We also note your complementary remark on the code's main README which we will of course provide in the final version of the artifact. As suggested we will also add further implementation details on the fusion in the appendix.
>
> Thank you again for your valuable feedback. Please let us know if you have further questions.
>
> **Reference**
>
> [1] Z. Huang et al. TinyChirp: Bird Song Recognition Using TinyML Models on Low-power Wireless Acoustic Sensors, in Proceedings of the IEEE International Symposium on the Internet of Sounds. Erlangen, Germany, September 2024.

---

### Official Review · Reviewer_eHEL · 2025-03-19

**Overall Recommendation:** 4

**Summary:**

The paper presents msf-CNN, a novel approach with open-source code to optimize convolutional neural network (CNN) inference on microcontrollers (MCUs) by employing multi-stage fusion techniques. Motivated by TinyML’s stringent memory and computational constraints, it reformulates the fusion configuration search as a graph-based shortest-path problem with constraint optimization, aiming to minimize peak RAM usage or computational cost. Key findings include substantial memory savings (up to 87% reduction compared to prior methods like MCUNetV2 and StreamNet), achieved through a directed acyclic graph (DAG) representation, a pruning strategy reducing complexity from O(2^(N-2)) to O(N^2), and iterative optimizations for global pooling and dense layers. Experimental results validate the approach across ARM Cortex-M, RISC-V, and ESP32 MCUs, showcasing flexibility for diverse IoT scenarios.

**Claims And Evidence:**

The claims—significant RAM reduction, flexible trade-offs, and efficient optimization—are robustly supported by evidence. Analytical results (Table 1) and experiments (Tables 3-5) show up to 87% RAM savings and consistent performance under constraints, with clear comparisons to MCUNetV2 and StreamNet. No unsupported claims were identified; the evidence is convincing and well-presented.

**Essential References Not Discussed:**

No critical omissions were noted. The paper cites key works (e.g., MCUNetV2, StreamNet) and foundational fusion studies, providing sufficient context. No recent, essential breakthroughs appear overlooked.

**Experimental Designs Or Analyses:**

I reviewed the experiments in Section 8 (Tables 3-5), testing minimal RAM usage, RAM budget, and compute cost limits. The design—spanning multiple MCU architectures and comparing to MCUNetV2 and StreamNet—is solid, with results corroborating analytical predictions (Section 6.3). The validation is thorough, and no issues arose.

**Methods And Evaluation Criteria:**

The methods—graph-based fusion optimization, iterative layer computation, and H-cache usage—are well-suited for TinyML’s resource-limited context. The evaluation criteria, measuring peak RAM (kB) and latency (ms) on diverse MCU boards (Table 2), are practical and relevant. Benchmarks like MobileNetV2 and MCUNet variants are apt choices, reflecting real-world AIoT needs.

**Other Comments Or Suggestions:**

None

**Other Strengths And Weaknesses:**

Strengths:

The conversion of fusion optimization into a shortest-path problem using inverted dataflow graphs is innovative, leveraging efficient graph algorithms for practical results.
The detailed microTVM implementation and validation across MCU architectures (up to 87% RAM reduction) highlight its real-world utility.
The flexibility to tune fusion for varying IoT resource profiles is a significant advantage.
Clarity and originality shine through, combining existing ideas creatively for TinyML.

Weaknesses & Suggestions:

Increased Computation Latency: The 2× to 5× latency increase for minimal RAM settings (Table 4) may limit real-time use. Exploring latency optimization could broaden applicability.
Hardware-Specific Considerations: Performance varies across architectures (e.g., Xtensa vs. RISC-V, Table 4); more insights into hardware-specific tuning would enhance portability.
Parameter and Architecture Exploration: The fixed output elements and H-cache focus limit the search space. Expanding to dynamic parameters or other architectures (e.g., transformers, RNNs) could amplify impact.

**Questions For Authors:**

None. The paper is clear, and limitations are acknowledged as future work, not requiring responses to shift my evaluation.

**Relation To Broader Scientific Literature:**

The paper extends prior fusion work (e.g., Alwani et al., 2016; Lin et al., 2021) by introducing multi-stage fusion and graph-based optimization, aligning with TinyML advancements (e.g., Lin et al., 2020). It builds on dataflow graph concepts (TensorFlow, PyTorch) and complements memory-efficient techniques like TinyEngine (Lin et al., 2021), enhancing their MCU applicability.

**Theoretical Claims:**

I verified the optimization problems (P1, P2) and MAC analysis (Appendix A). The conversion to a shortest-path problem using inverted dataflow graphs is theoretically sound, and equations (e.g., Eq. 5, 16, 17) for cache size and MAC counts are correctly formulated. No errors were found.

---

> ### Author Rebuttal · Authors · 2025-04-01
>
> Thank you so much for the supportive comments and for your valuable suggestions for future work in various directions. We are incidentally working on exploring hardware-specific tuning, leveraging more advanced caching techniques and covering architectures other than CNN.

---

### Official Review · Reviewer_FvtT · 2025-03-23

**Overall Recommendation:** 2

**Summary:**

This paper present msf-CNN, by leveraging traditional DAG and kernel fusion techniques, msf-cnn achieves 50% less peak memory usage while achieving similar inference performance. The paper logic flow is clear, but lack of novelty, and some experiment results baseline are not strong.

## update after rebuttal

After careful review of authors comments, I still want to remain my review and score. Thanks

**Claims And Evidence:**

No. For example, the most important claim that the paper repeatedly highlighted as performance gain is not clear claim. e.g., "msf-CNN can achieve inference with less than 50% the peak RAM usage state-of-the-art". What does it mean by state-of-the-art, in which dimension latency? throughput? or accuracy?

**Essential References Not Discussed:**

TorchDynamo, TorchCompile, ExecuTorch (https://pytorch.org/executorch/stable/index.html)

**Experimental Designs Or Analyses:**

baseline too weak.

**Methods And Evaluation Criteria:**

Methods: the paper use very standard graph optization and kernel fusion techniques in ML compiler level. For example, use DAG to describe computation flow, and leveraging kernel fusion to reduce peak memory i/o or reduce recomputation.

Optimizing for either low memory usage or less compute make sense. However, the baseline that this paper compared with is too weak. It compared against pure vanilla execution of sequential kernels without any basic kernel fusion techniques.

**Other Comments Or Suggestions:**

In related work section 10, ". However, none of the above tools provide CNN fusion optimization mechanisms, in contrast to msf-CNN". I don't see it is true. For example, in tvm paper (https://homes.cs.washington.edu/~arvind/papers/tvm.pdf), section3, operator fusion part, it clearly side conv2d can be fused with other element-wise kernels.

**Other Strengths And Weaknesses:**

Strength:
1. logic flow is very clear and easy to follow
2. theoretical analysis are easy to follow

Weaknesses:
1. methods used are very basic and standard practice in main-stream AI inference optimization, such as describe compute flow as DAG (same as torch.fx),   kernel fusion ( torch.compile,  torchdynamo), deployment to edge device ( ExecuTorch).
2. In experimental design, vanilla baseline is too weak for comparison purpose. Any inference systems nowadays includes some level of automatic kernel fusion techniques and kernel dimension fine-tuning. Therefore the vanilla baseline of sequencial execution of every kernel is too weak to compare with.

**Questions For Authors:**

Over the paper is trying to optimize for lower memory usage or less computation cost on edge device, which is a promising direction. I have several questions:

1. how does this work compared with main-stream kernel fusion techniques such as torch.compile, torchDynamo, torch fx.graph, ExecuTorch?

2. For inference reducing peak mem usage or less computation, how does this work different from unsloth (https://github.com/unslothai/unsloth), liger (https://github.com/linkedin/Liger-Kernel) and similar work?

3. kernel fusion based on computation graph optimization is well defined area. As mentioned in this paper, tvm or micro-tvm already fully support such feature (also with automatic fusion schema). Any novel contribution made in this paper?

**Relation To Broader Scientific Literature:**

This contribution can be helpful to reducing peak memory usage or reducing recomputation cost in CNN inference on edge devices.

**Theoretical Claims:**

yes

---

> ### Author Rebuttal · Authors · 2025-04-01
>
> Thank you very much for your comments pointing out that a reader might potentially confuse **multi-stage fusion** (msf-CNN, our approach) on the one hand, and on the other hand traditional **kernel fusion** techniques. These two approaches are orthogonal and can be applied concurrently for maximum benefit.
>
> While kernel fusion optimizes computation overhead, msf-CNN targets instead memory efficiency, the latter being the critical first hurdle on small edge devices. As such, we could disambiguate this further in the related work section:
>
> - **Kernel fusion** [1-3] focuses primarily on reducing redundant data movements between GPU and RAM by combining multiple primitive operators (e.g. Batch Normalization, ReLU, Softmax, etc.) with a primary, memory-bound operator (e.g. conv, pooling) into a single kernel. While kernel fusion improves compute latency and data throughput, it does not address the fundamental memory usage issue that arises when processing multiple primary operators sequentially.
>
> - **multi-stage fusion (msf-CNN, our approach)** extends the idea of *patch-based fusion* [4, 5]. More specifically, msf-CNN:
>     - Fuses multiple layers (i.e. primary operators like convolution and pooling) into a single computational stage.
>     - Implements _patch-based partial computation_, which drastically reduces peak memory usage by processing input data in smaller patches while maintaining accuracy.
>     - Introduces a compute-memory trade-off mechanism that allows users to prioritize either memory consumption or computational efficiency based on their deployment constraints.
>
> This makes msf-CNN fundamentally different from traditional kernel fusion techniques.
>
> To the best of our knowledge, the closest related works to msf-CNN are StreamNet and MCUNetv2, which have been the state-of-the-art for patch-based fusion on microcontrollers so far. Compared to this state of the art, based on our measurements, msf-CNN achieves up to 50% reduction of peak RAM usage in model inference, for the same inference accuracy, which thus enables more models to fit smaller devices.
>
> We provide below our answers to the other questions you raised.
>
> **[Q1. how does this work compared with main-stream kernel fusion techniques such as torch.compile, torchDynamo, torch fx.graph, ExecuTorch?]**
>
> While mainstream tools like TorchDynamo and ExecuTorch implement kernel fusion to optimize compute latency, msf-CNN introduces a novel layer-level fusion strategy that specifically targets memory efficiency. Our approach can be applied in conjunction with existing kernel fusion techniques to achieve both latency and memory optimizations. Without lacking the generality, we leave this for future development.
>
> **[Q2. how does this work different from unsloth, liger and similar work?]**
>
> unsloth and Liger-kernel focus on optimizing large language models (LLMs) for fine-tuning and (post-)training scenarios, primarily targeting GPU-based systems. Their optimizations, such as LoRA and FP8/4 quantization, are designed for LLM-specific workloads and not targets on general-purpose CNN inference on edge devices. In contrast, msf-CNN is designed for general-purpose CNN inference on tiny edge devices with limited resources.
>
> **[Q3. kernel fusion based on computation graph optimization is well defined area...Any novel contribution made in this paper?]**
>
> The relation between kernel fusion and our method is explained above. Moreover, the key novel contributions of our work include:
> - A compute-memory trade-off framework with efficient optimizer that allows users to optimize for specific resource constraints.
> - A patch-based fusion mechanism that enables layer-level fusion and significant memory savings.
> - A general-purpose solution for CNN inference on edge devices, which is distinct from existing tools focused on LLM optimization.
> - An open source framework enabling ultra-low RAM footprint of neural network inference.
>
> Thank you again for your valuable feedback. Please let us know if you have further questions.
>
> **Reference**
>
> [1] Wang, et al. "Kernel fusion: An effective method for better power efficiency on multithreaded GPU." 2010 IEEE/ACM Int'l Conference on Green Computing and Communications & Int'l Conference on Cyber, Physical and Social Computing. IEEE, 2010.
>
> [2] Niu, Wei, et al. "Dnnfusion: accelerating deep neural networks execution with advanced operator fusion." Proceedings of the 42nd ACM SIGPLAN International Conference on Programming Language Design and Implementation. 2021.
>
> [3] Zhao, Jie, et al. "Apollo: Automatic partition-based operator fusion through layer by layer optimization." Proceedings of Machine Learning and Systems 4 (2022): 1-19.
>
> [4] Alwani, Manoj, et al. "Fused-layer CNN accelerators." 2016 49th Annual IEEE/ACM MICRO. IEEE, 2016.
>
> [5] Mei, Linyan, et al. "Defines: Enabling fast exploration of the depth-first scheduling space for dnn accelerators through analytical modeling." 2023 IEEE HPCA. IEEE, 2023.

---

### Decision · Program_Chairs · 2025-05-01

**Decision:**

Reject

**Comment:**

After rebuttal, multiple reviewers have expressed concerns about limited novelty (straightforward objective -> come up with heuristics to solve -> reasonable performance) and asked for more comparison against previous methods (TorchDynamo, TorchCompile, ExecuTorch, etc). While these previous methods are not focused on peak memory reduction (and thus is not optimizing the objective in Sec. 4), they also rely on heuristics to do kernel fusion and deserve some comparison. The paper could benefit from another round of revision.